# Proteomic and Genomic Changes in Tau Protein, Which Are Associated with Alzheimer’s Disease after Ischemia-Reperfusion Brain Injury

**DOI:** 10.3390/ijms21030892

**Published:** 2020-01-30

**Authors:** Marzena Ułamek-Kozioł, Stanisław Jerzy Czuczwar, Sławomir Januszewski, Ryszard Pluta

**Affiliations:** 1Laboratory of Ischemic and Neurodegenerative Brain Research, Mossakowski Medical Research Centre, Polish Academy of Sciences, 02-106 Warsaw, Poland; mulamek@imdik.pan.pl (M.U.-K.); sjanuszewski@imdik.pan.pl (S.J.); 2Department of Pathophysiology, Medical University of Lublin, 20-090 Lublin, Poland; czuczwarsj@yahoo.com

**Keywords:** Brain ischemia, stroke, neuronal death, tau protein, gene expression, dementia, neurodegeneration, human, animals

## Abstract

Recent evidence suggests that transient ischemia of the brain with reperfusion in humans and animals is associated with the neuronal accumulation of neurotoxic molecules associated with Alzheimer’s disease, such as all parts of the amyloid protein precursor and modified tau protein. Pathological changes in the amyloid protein precursor and tau protein at the protein and gene level due to ischemia may lead to dementia of the Alzheimer’s disease type after ischemic brain injury. Some studies have demonstrated increased tau protein immunoreactivity in neuronal cells after brain ischemia-reperfusion injury. Recent research has presented many new tau protein functions, such as neural activity control, iron export, protection of genomic DNA integrity, neurogenesis and long-term depression. This review discusses the potential mechanisms of tau protein in the brain after ischemia, including oxidative stress, apoptosis, autophagy, excitotoxicity, neurological inflammation, endothelium, angiogenesis and mitochondrial dysfunction. In addition, attention was paid to the role of tau protein in damage to the neurovascular unit. Tau protein may be at the intersection of many regulatory mechanisms in the event of major neuropathological changes in ischemic stroke. Data show that brain ischemia activates neuronal changes and death in the hippocampus in a manner dependent on tau protein, thus determining a new and important way to regulate the survival and/or death of post-ischemic neurons. Meanwhile, the association between tau protein and ischemic stroke has not been well discussed. In this review, we aim to update the knowledge about the proteomic and genomic changes in tau protein following ischemia-reperfusion injury and the connection between dysfunctional tau protein and ischemic stroke pathology. Finally we present the positive correlation between tau protein dysfunction and the development of sporadic Alzheimer’s disease type of neurodegeneration.

## 1. Introduction

Ischemic brain injury in the human clinic is the second leading cause of death and the third leading cause of physical disability, and may soon become the main cause of Alzheimer’s disease type dementia, and, currently, brain ischemia is proposed to be a risk factor for developing Alzheimer’s disease [1,2,3,4,5,6]. The number of patients who have survived an ischemic stroke with a severe neurological deficit has now reached 33 million, and their number will increase to 77 million by 2030 [3]. Neurological deficits after stroke in people have a tendency to improve to a greater or lesser extent. However, for reasons unknown, the cognitive deficit is gradually deteriorating, leading to the development of dementia of the Alzheimer’s disease type. At present, human stroke has a heavy burden on global public health and social care as well as clinical practice.

Human and animal brain ischemia causes the production of β-amyloid peptide accumulation and further impairs the removal of the neurotoxic β-amyloid peptide from the extra- and intracellular space of the brain [7,8,9,10,11,12,13,14]. Additional evidence suggests that ischemic brain damage in humans and animals may contribute to tau protein dysfunction, especially in neurons [15,16]. The dangerous and regular damage of the brain after ischemia-reperfusion is the progressive and delayed dementia of the Alzheimer’s disease type [2,4,6,17,18,19,20,21,22,23,24,25]. Previous brain damage associated with ischemia and reperfusion may further increase the likelihood of developing dementia associated with Alzheimer’s disease, increasing the extent of post-ischemic changes, through the proteomic and genomic cascade associated with Alzheimer’s disease [26,27,28,29,30,31,32,33,34,35,36,37]. Based on the above observations, it has been suggested that the history of cerebral ischemia in humans and animals is associated with the subsequent development of sporadic Alzheimer’s disease [1,7,12,15,16,17,26,27,28,29,30,31,32,33,34,35,36,38,39,40,41,42,43,44,45,46]. In this review, we focus first on identifying the response of the tau protein gene and its product to ischemia-reperfusion brain damage. Secondly, we will present the contribution of tau protein after ischemia to the development of sporadic Alzheimer’s disease type of neurodegeneration, focusing on both changes in its structure and the expression of its gene after brain ischemia insult.

## 2. Structure, Physiological and Pathological Activity of Tau Protein

Tau protein was first isolated and named in 1975 for its ability to induce tubule formation, and was mostly segregated into neuronal axons. Tau protein can be also detected in oligodendrocytes. Besides the nervous system, tau protein was also found in many other tissues, such as heart, lung, kidney, and testis, but less abundant. The tau protein is a phospho protein and its action depends on the level of its phosphorylation. The tau protein is naturally unfolded with low secondary structure content. Tau protein is composed of four regions: an N-terminal projection region, a proline-rich domain, a microtubule-binding domain, and a C-terminal region (Figure 1) [47]. Six isoforms of tau protein have been found in human adult brains; they are expressed by alternative splicing around the N-terminal projection region and microtubule-binding domain. Tau protein is mainly expressed in the brain, it has six isoforms produced by alternative mRNA splicing of *microtubule-associated tau protein* gene which comprises 16 exons on chromosome 17q21.31. The primary physiological function of tau protein is to stabilize microtubule networks within neurons, whereas the hyperphosphorylated condition will significantly reduce its biological activity. The main physiological tau protein function in the cell is regulating microtubule structure and dynamics by binding to microtubules, it has been also proven in cell-free conditions. Furthermore, the dynamic microtubule network provided by tau protein is important to the proper migration of new neurons, and severe reduction of adult neurogenesis was found in tau protein knockout mice [47]. The tau protein’s functions are regulated by a complex array of post-translational modifications, such as glycation, phosphorylation, isomerization, acetylation, sumoylation, nitration, O-GlcNAcylation, and truncation—these suggest that tau protein plays opposite roles in physiology and pathology [47]. According to previous observations, the kinds of dysfunctional tau protein are different in diverse brain ischemia models, such as neurofibrillary tangle formation, hyper-phosphorylation, dephosphorylation, and re-phosphorylation (Table 1). The hyper-phosphorylated state is the particularly pathological condition of tau protein in post-ischemic brains. It decreases the affinity of tau protein for the microtubules by disrupting the binding balance [47]. The tau protein contains a large amount of serine and threonine residues, which are potential phosphorylation sites, and the phosphorylation state, which is controlled by the balance of kinase and phosphatase activity, affects the affinity of microtubule binding. As the tau protein is phosphorylated by kinases involved in various transduction signaling pathways, its degree of phosphorylation controls its binding to microtubules, affecting the dynamics of microtubule assembly necessary for axon growth and neurite plasticity [48]. Hyperphosphorylated tau protein does not bind or stabilize microtubules, while fully dephosphorylated tau protein binds to microtubules with high affinity. Brain ischemia damages the neuronal cytoskeleton both by promoting its proteolysis and by affecting the activity of kinases and phosphatases [49]. Therefore, the physiological activity of the tau protein preferentially affects the development of microtubules and their stabilization by phosphorylation. Microtubules are involved in maintaining the structure of neurons and creating axonal and dendritic processes, and play an important role in vesicular axonal transport and signal transduction. Modifications of tau protein phosphorylation may alter its circulation between the axon and the cell body and affect susceptibility to proteolysis, affect microtubule stability and may contribute to disrupting axonal transport, but also facilitate neurite plasticity in the regenerative response [48]. Another study showed that the tau protein alone can reduce the transport of the amyloid protein precursor from the body of the neuron to axons and dendrites, leading to the storage of the amyloid protein precursor in the body of the nerve cell [50]. Current research presents numerous new functions of tau protein, such as neural activity control, iron export, protection of genomic DNA integrity, neurogenesis and long-term depression [16]. 

### 2.1. Tau Protein in Post-Ischemic Brain

Initial tau protein staining was presented in both neural and glial cells in the hippocampus, cortex and thalamus in both experimental and human brain ischemia [59,64,65,66,67,68,69,70,71]. Tau protein was also observed in microglia after focal ischemia of the brain in ischemic penumbra [57,71]. The data presented indicate that some neurons show changes in the tau protein after ischemia-reperfusion brain injury [67], which may be related to the main neuropathological stage of ischemic processes in these cells [69].

### 2.2. Tau Protein in the Blood After Brain Ischemia and Ischemic Blood-Brain Barrier

An increased level of amyloid in the blood after an ischemic episode [72,73] can indirectly affect changes in tau protein in the blood, representing an automated link between the accumulated amyloid and tau protein pathology after ischemic blood–brain barrier failure [74]. In addition, both oxidative stress [75] and neuroinflammation [76] induced by the permeability of the blood–brain barrier may initiate phosphorylation of tau protein and the development of neurofibrillary tangles after brain injury as a result of ischemia-reperfusion (Table 1) [16,49,51,54,55,77]. Tau protein accumulated in the blood after a ischemia-reperfusion brain episode [78,79] can cross the ischemic blood–brain barrier and tau protein originating from serum can cause a stronger tau protein pathology in the brain parenchyma [80]. Ischemia-reperfusion brain injury with ischemic insufficiency of the blood–brain barrier [7,81,82,83,84,85] initiates tau protein phosphorylation [53,54,56,57,77], and phosphorylated tau protein may cause damage to the blood–brain barrier, leading to harmful feedback reactions [74]. The permeability of the blood–brain barrier may exacerbate neuropathology through the tau protein from blood in brain damage as a result of ischemia-reperfusion by increasing its level in brain tissue, which suggests that the ischemic-reperfusion episode of the brain may play an important role in the growth of the blood tau protein level [78,79,80].

### 2.3. Dysregulation of the Tau Protein Gene After Brain Ischemia

A recent report indicated the relationship between hippocampal CA1 region neuron damage and the expression of the *tau protein* gene after 10 min experimental global brain ischemia due to cardiac arrest, with recirculation of 2, 7 and 30 days [15]. In the neurons of the CA1 area, the *tau protein* gene expression increased to a maximum of 3-fold change on the second day after brain ischemia [15]. On the seventh day of reperfusion after the ischemic episode, gene expression ranged from 0.2 to −0.5-fold change [15]. On the thirtieth day of recirculation after brain ischemia, the expression of the *tau protein* gene was below the control values [15]. The statistical significance of the changes in the neuronal gene expression of the tau protein after brain ischemia-reperfusion injury in rats was between 2 and 7, and 2 and 30 days of recirculation [15].

In the CA3 region of the hippocampus, the expression of the *tau protein* gene after ischemic injury with a survival of 2 days was lower than the control values and higher than the control values on days 7–30. On the second day after ischemia, the minimum was a −0.6-fold change and the maximum was a −0.001-fold change, with a median −0.2-fold change [86]. On the seventh day after ischemia, the minimum was a 0.1-fold change and the maximum was a 0.6-fold change, with a median 0.2-fold change [86]. On the thirtieth day after ischemia, the minimum was a 0.03-fold change and the maximum was a 0.34-fold change, with a median 0.18-fold change [86]. The changes were statistically significant between 2 and 7 days and between 2 and 30 days after ischemia [86].

Data show that brain ischemia activates neuronal changes and death in the hippocampus in a manner dependent on tau protein, thus determining a new and important way to regulate the survival and/or death of post-ischemic neurons.

### 2.4. Phosphorylation of Tau Protein After Brain Ischemia

In some studies, after an experimental focal and global ischemia-reperfusion episode of the brain, dephosphorylation of tau protein was demonstrated (Table 1) [48,62,66,67]. After transient complete cerebral ischemia with recirculation due to cardiac arrest, the tau protein was gradually re-phosphorylated (Table 1) [62]. Transient local brain ischemia in rats with recirculation induced a site-specific hyperphosphorylation of the tau protein (Table 1) [54]. During the death of neurons in the CA1 region of the hippocampus after transient cerebral ischemia in the gerbil, hyperphosphorylation of serine 199/202 tau protein was regulated by GSK3, MAP kinase and CDK5 activity (Table 1) [58]. In addition, it was observed that the microglial tau protein is phosphorylated after ischemic brain damage in humans (Table 1) [71]. Current research indicates that after transient focal and global ischemia of the brain with reperfusion, modifications of the hyperphosphorylation of tau protein are similar to those occurring in Alzheimer’s disease and predominate in cortical neurons and are accompanied by apoptosis (Figure 2) [49,55,57,87]. The above data indicate that, after ischemia-reperfusion brain injury, neural apoptosis is directly related to the hyperphosphorylation of the tau protein. Khan et al. [52] showed an increase in the production of paired helical tau protein filaments after global cerebral ischemia in mice. Wen et al. [49,55] provided evidence that transient brain injury due to ischemia and reperfusion was involved in Alzheimer’s disease-like neurofibrillary tangle generation in female rats after local cerebral ischemia (Table 1). The formation of neurofibrillary tangles was observed after focal ischemia-reperfusion injury of the brain on the side of massive cerebral infarction in humans (Table 1) [54]. In addition, the combination of total brain ischemia with hyperhomocysteinemia in rats led to enormous neuronal changes in the hippocampus and cortex caused by hyperphosphorylated tau protein (Table 1) [56]. The above study reported a 695-fold increase in hyperphosphorylated tau protein-positive neurons in the ischemic brain compared to the control [56]. As an endpoint, the tau protein, a fundamental feature of Alzheimer’s disease, aggravates brain tissue damage in transient experimental brain ischemia through tau protein excitotoxicity (Figure 2) [63,88] and tau protein-mediated iron export [89].

## 3. Tau Protein Hyperphosphorylation Renders Cells More Resistant to Apoptosis?

Hyperphosphorylated tau protein, most likely by competitive inhibition of GSK-3 phosphorylation of β-catenin, facilitates the action of β-catenin and other proteins, thus inhibiting the apoptosis pathway [90,91,92,93,94]. It was also shown that neurons with dephosphorylated tau protein were more susceptible to apoptosis [60]. The involvement of tau protein in the neuron viability was also observed in the cerebellar granule neurons [95]. These studies suggest that hyperphosphorylated tau protein may lead to the breakdown of acute apoptosis in neurons. Because adult neurons are rarely replenished, the failed apoptosis induced by the phosphorylation of the tau protein may be one of the evolving mechanisms that may allow neurons to survive an apoptotic attack and wait for a chance of self-repair. Although hyperphosphorylation of the tau protein can cause neuron escape from the apoptotic pathway and thus prevent the rapid loss of many neurons by the brain, nerve cells with hyperphosphorylated tau protein are nevertheless “sick” (Figure 2 and Figure 3) and are no longer competent for normal physiological functions such as promoting microtubule assembly and maintaining normal axonal transport [96]. In addition, the prolonged survival time of these “sick” neuronal cells makes them less resistant to environmental influences, and also allows them to develop tangles from hyperphosphorylated tau protein (Table 1). Hyperphosphorylation of the tau protein leads to slow but progressive retrograde degeneration of neurons. Therefore, modulation of phosphorylation of tau protein at various stages of Alzheimer’s disease and related tauopathies offers promising ways to save neurons from degeneration.

## 4. Hyperphosphorylated Tau Protein Good or Bad?

The biological activity of the tau protein is regulated by its degree of phosphorylation; both hypo- and hyperphosphorylation [16,97]. The harmful effect of tau protein is largely deduced from the fact that hyperphosphorylated tau protein is accidentally present in degenerate neurons in several tauopathies, and in particular in Alzheimer’s disease and brain ischemia (Table 1). However, these correlations are not enough to conclude that hyperphosphorylated tau protein is the harbinger of cell death in Alzheimer’s disease. Instead, some recent studies show that hyperphosphorylation of tau protein can be protective, especially when cells are exposed to acute injuries [96]. Quantitative analysis of neuronal loss as a function of disease duration revealed that the CA1 hippocampal neurons carrying neurofibrillary tangles can survive for decades [98]. In transgenic mouse models expressing human tau protein, the presence of tau protein filaments did not correlate directly with the death of individual neurons [91,92,93,94,96] and, furthermore, formation of tau protein filaments seems neuroprotective [99,100]. These data suggest that the aggregation of hyperphosphorylated tau protein in the adult mammalian brain may be associated with neuroprotective mechanisms. There has been some controversy regarding the toxicity of the polymerized tau protein [97]. Some data show that tau protein aggregation is toxic to cells [101,102]. Conversely, some studies have shown that the polymerization of tau protein is not associated with toxic effects on cells. It has been found that the polymerized tau protein loses its biological activity with respect to binding to microtubules, while dephosphorylation of aberrantly hyperphosphorylated tau protein and paired helical filaments transforms them into a normal protein, restoring significantly the biological activity of the tau protein [103,104]. The reduction in the microtubule density in pyramidal neurons in the brains of Alzheimer’s disease patients is not associated with the presence of paired helical filaments [105]. It has been found that the removal of the pattern-breaking sequences in microtubule binding repeats results in immediate aggregation of the tau protein and toxicity, but toxicity appeared without the requirement of fibril formation [106]. In transgenic P301L mice, neuronal death in the CA1 hippocampus region was prevented when suppressing the expression of the mutant *tau protein* gene and improvement in memory was observed without decreasing the number of neurofibrillary tangles [107,108]. Chronic treatment of transgenic P301L mice with ERK2 inhibitor resulted in a significant reduction in hyperphosphorylated tau protein and prevented motor impairment, but the number of neurofibrillary tangles did not decrease in the successfully treated group [109]. The formation of tau protein aggregates abolished the toxicity of soluble phosphorylated tau protein [99]. From these observations, it appears that the formation of paired helical filaments/neurofibrillary tangles from soluble hyperphosphorylated tau protein in neurons is a defense mechanism by which neurons seek to reduce the toxic activity of soluble hyperphosphorylated tau protein.

## 5. Discussion

This review shows the response of the *tau protein* gene and its products to brain ischemia with recirculation (Figure 2 and Figure 3). The data revealed that after ischemic brain injury, the overexpression of the *tau protein* gene 2 days after ischemic episode began and correlated with a huge increase in plasma tau protein after ischemic injury [78,79] and extracellular space after brain injury [110], as well as with the hyperphosphorylation of tau protein in ischemic brain tissue (Table 1) [49,54,57,85,87]. The increased expression of the tau protein gene was parallel to the onset of delayed neuronal death in the hippocampus after ischemia [11,12,39]. A rise in tau protein levels in the brain and blood [77,79] was associated with a similar increase in the β-amyloid peptide level in the brain and serum after ischemia (Figure 3) [72,73], and this observation forecasted a worse clinical outcome. The increase in *tau protein* gene expression induced by ischemia at the onset of neuronal death in the hippocampus is parallel to the overexpression of the *caspase 3* gene, which plays a role in neuronal death (Figure 3) [35]. The processes by which both the tau protein and caspase 3 kill the neurons of the hippocampus are not completely understood. Caspase ultimately cuts the tau protein into shorter forms [16]. Notably, studies have shown that activated caspase positively correlates with increased levels of truncated tau protein and the formation of neurofibrillary tangles [16]. In addition, cognitive deficits are negatively correlated with the level of tau protein shortened by caspase 3 [16]. The data suggest that when the tau protein undergoes ischemic translation, its hyperphosphorylation increases, which means that the hyperphosphorylation of the tau protein is driven by the substrate and the transcription levels are identical to the protein levels [49,55]. Other studies have shown elevated levels of Cdk5 in rats exposed to focal transient cerebral ischemia, confirming the above observations [49]. Enhancing the hyperphosphorylation of the tau protein level may be a consequence of the increased translation of the tau protein and the inhibition of tau protein degradation and/or blocked clearance. The inhibition of degradation is strongly supported by a reduction in the level of the autophagy gene expression during the onset of neuronal death in the hippocampus [35]. The immunocytochemical studies demonstrated that intraneuronal tau protein proteolysis is a sensitive, early marker of focal ischemic injury in the brain. The double labeled immunofluorescence experiments suggested that proteolysis of tau protein coincides with calpain activation. It was concluded that focal brain ischemia is associated with early microtubular proteolysis caused by calpain [111]. Data show that brain ischemia activates neuronal changes and death in the hippocampus in a manner dependent on tau protein, thus determining a new and important way to regulate the survival and/or death of post-ischemic neurons. Triggered pathological changes such as oxidative stress, apoptosis, autophagy, excitotoxicity, inflammation, endothelium, angiogenesis, and mitochondrial dysfunction of tau protein determine its potential regulatory mechanisms in ischemic stroke (Figure 2).

The presented facts confirm the opinion that brain ischemia with reperfusion plays a key role in the dysfunction of the tau protein in the brain and blood after ischemia. The expression of the *tau protein* gene and its protein level in brain tissue and plasma, which are increased after ischemic brain injury [15,78,79], are involved together with a parallel generated amyloid (Figure 3) in the development of neuropathology characteristic of Alzheimer’s disease after ischemia. One study proved that the regional redistribution of tau protein from the neuropil to neuronal perikarya in post-ischemic stroke model was thought to share similarity with that occurring in Alzheimer’s disease [57]. It is highly likely that the modified tau protein additionally enhances ischemic neuronal damage after ischemia (Figure 3). The above data allow us to understand the acute and chronic processes during neuronal death and the development of slow and progressive brain atrophy after ischemic damage with dementia of the Alzheimer’s disease phenotype [12,22,112,113]. After brain ischemia injury, an increase in blood tau protein levels was observed in patients [78,79]. Increased plasma tau protein levels correlated negatively with the clinical outcome after ischemic brain injury, which, in turn, reflected the severity of the ischemic stroke [78,79]. We can conclude that proteomic and genomic changes in tau protein, which are associated with Alzheimer’s disease pathology, contribute to the neurodegeneration of the brain after ischemia with the development of the type of Alzheimer’s disease dementia [12,22,112,113]. In the brain after ischemia with reperfusion, ischemia seems to favor the development of irreversible neurodegeneration of the Alzheimer’s disease type with neuronal death [11], neuroinflammation [76], white matter changes, general brain atrophy, amyloid accumulation [7,45] and dysfunctional tau protein (Figure 2 and Figure 3) [15,16]. Although significant progress has recently been made in studying the pathogenicity of tau protein after brain injury due to ischemia and reperfusion, the key mechanisms/pathways involved in irreversible brain neurodegeneration induced by tau protein after ischemia are still unknown. It has also been shown that ischemia with reperfusion injures the brain, inducing neuronal death in the brain in a manner dependent on the tau protein (Figure 2 and Figure 3) [16], thus defining a new and important way to regulate the survival or death of neurons. The relationship between tau protein associated with Alzheimer’s disease and experimental cerebral ischemia and ischemic stroke in humans seems quite clear. The worldwide problem and the huge costs associated with human ischemic stroke clearly show that there is an urgent need to progress in the treatment of post-ischemic brain injury with irreversible consequences, such as the dementia of the Alzheimer’s disease phenotype.

Although the role of ischemia in the hyperphosphorylation of tau protein is generally complex and requires further research, and tau protein is a relatively undervalued factor in ischemic stroke, we have reason to believe that determining the role of tau protein in cerebral ischemia may help to understand the basis for developing a new target for treatment of ischemic stroke (Table 1). It seems that these data confirm that the regulation of tau protein phosphorylation can be considered as a potential new therapeutic target after ischemic stroke.

According to observations from earlier and newer studies, it can be concluded that transient global and focal ischemia brain damage affects the modification of the tau protein at both the protein and gene level, leading to tau protein deposition as paired helical filaments, neurofibrillary tangle-like and neurofibrillary tangles in the brain (Table 1) [16]. The conclusions drawn from the analysis of ischemia-triggered Alzheimer’s disease-related tau protein and its gene in the brain, which are part of the cause of neuronal death by generation of neurofibrillary tangle-like and/or neurofibrillary tangles, are crucial for the improvement of therapy of irreversible post-ischemic neurodegeneration. Because the accumulation of amyloid and tau protein is not the cause of Alzheimer’s disease pathogenesis, as found in the NIA-AA Research Framework: towards the biological definition of Alzheimer’s disease [114], it is understandable that advanced study is necessary in this area. Ultimately, the experimental models of ischemia-reperfusion brain damage used in the study of Alzheimer’s disease seem to be a useful new approach to clarifying the role of folding proteins and their genes in neurodegenerative diseases such as cerebral ischemia and sporadic Alzheimer’s disease [15,16,27,28,29,30,34,35,115,116,117,118,119].

## Figures and Tables

**Figure 1 ijms-21-00892-f001:**
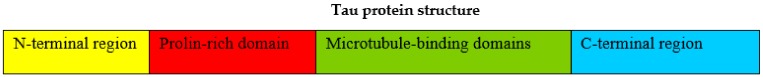
Structure of tau protein: N-terminal region, prolin-rich domain, microtubule-binding domains and C-terminal region. 1–441 number of amino acids.

**Figure 2 ijms-21-00892-f002:**
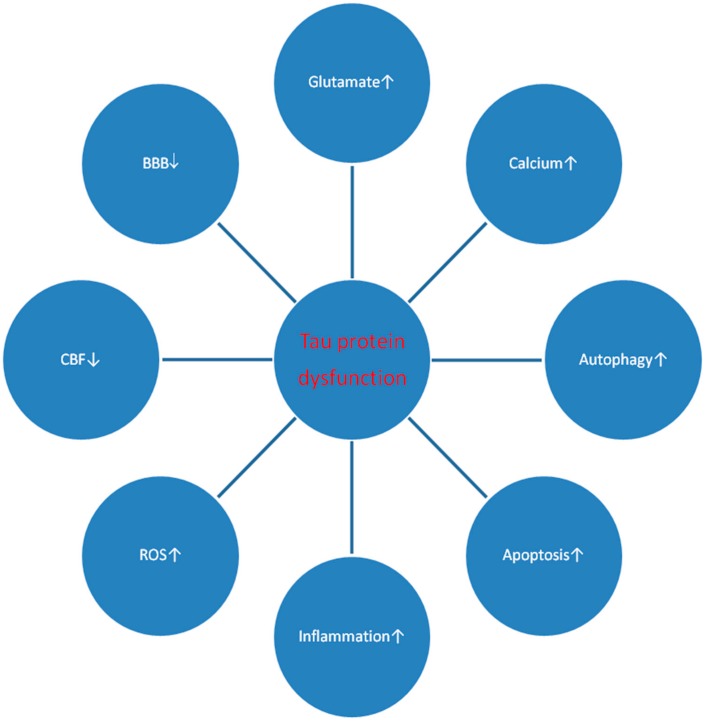
Potential regulatory mechanisms of dysfunctional tau protein in post-ischemic brain neuropathology. ↓ - decrease, ↑ - increase. BBB–blood–brain barrier, CBF–cerebral blood flow, ROS–reactive oxygen species.

**Figure 3 ijms-21-00892-f003:**
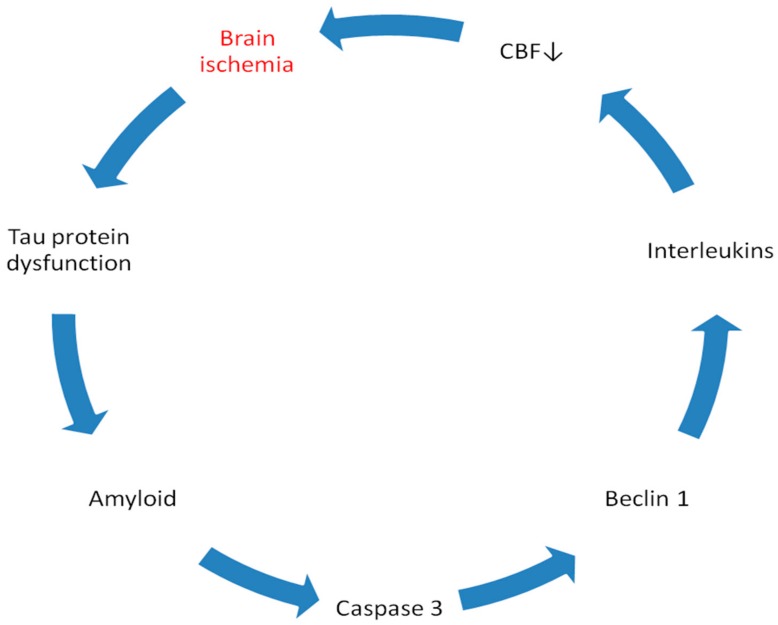
Cross talk between Alzheimer’s disease-associated proteins during post-ischemic brain injury. Beclin 1-protein associated with autophagy. CBF↓-decrease of cerebral blood flow.

**Table 1 ijms-21-00892-t001:** Different patterns of tau protein phosphorylation in post-ischemic brain.

Animal or Human	Kind of Ischemia	Time of Ischemia	Region of Brain	Tau Protein Changes	Tau Protein Phospho-Site	Effect of Tau Protein Changes	Ref.
Human	Ischemic stroke	Lack	Ischemic cortex	Neurofibrill-ary tangle	Tau 1	Final stage of tau changes	[51]
Mouse	Global ischemia	10,15,18 min.	Hippocampus, cortex	Paired helical filaments	Ps396, Ps404	Neuron death	[52]
Rat	Focal ischemia	1 h.	Ischemic cortex	Neurofibrill-ary tangle- like	P-396, P-404	Progression of ischemic changes	[49]
Mouse	Focal ischemia	90 min	Ischemic core	Hyperphos-phorylation	Ser262, Ser356	Involve- ment of asparagine endopepti- dase	[53]
Rat	Focal ischemia	1 h	Ischemic cortex	Hyperphos-phorylation	PT181, pS202, pT205, pT212, pS214, pT231, pS262, pS396, pS404, pS422	Destabiliza- tion of neuronal cytoskeletonand apoptosis	[54]
Rat	Focal ischemia	1 h	Ischemic cortex	Hyperphos-phorylation	Phospo-tau protein 202/205, 214, 396/404, 231.	Progression of ischemic changes	[55]
Rat	Global ischemia	15 min	Ischemic cortex	Hyperphos-phorylation	Ser202, Thr205	Oxidative stress, neuron, astrocyte damage.	[56]
Rat	Focal ischemia	90 min	Ischemic core	Hyperphos-phorylation	Asp421	Axonal changes	[57]
Gerbil	Forebrain ischemia	5 min	Hippocampus	Hyperphos-phorylation	Ser199, Ser202	Induction MAP kinase, CDK5, GSK3, neuronal damage	[58]
Human	Ischemic stroke	Lack	Ischemic cortex	Hyperphos-phorylation	Ser101	Microglia tau protein injury	[59]
Rat	Global ischemia	2,8 min	Cortex, hippocampus	Phosphory- lation, dephospho- rylation	Ser 396, 262, 202, Thr205	AMPK changes	[60]
Mouse	Focal ischemia + hypoxia	40 min	Ischemic core	Decrease in phosphoryla-tion	P301L	Accumula- tion of glutamate	[61]
Rat	Global ischemia	5,15 min	Neocortex, hippocampus, striatum	Dephospho-rylation	Ps396, Ps404	Changes in axonal transport	[48]
Dog	Global ischemia due to cardiac arrest	10 min	Cortex	Dephospho-rylation, rephospho- rylation	Ser262, Ser356	Neuronal changes	[62]
Mouse	Focal ischemia	90 min	Ischemic cortex	Tau protein -/- in mice	Lack	Reduce excitotoxici-ty	[63]

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
