# Peer review of "Proteomic and Genomic Changes in Tau Protein, Which Are Associated with Alzheimer’s Disease after Ischemia-Reperfusion Brain Injury"

_ijms, 2020, doi:10.3390/ijms21030892_

Round 1

Reviewer 1 Report

The review is exceptionally written and comprehensive with respect to how Tau protein is influenced in the brain in the context of ischemia. The authors draw firm conclusions backed by extensive evidence and the review does a good job of suggesting several hypotheses and topics that merit further study.

Author Response

Reviewer 1. The review is exceptionally written and comprehensive with respect to how Tau protein is influenced in the brain in the context of ischemia. The authors draw firm conclusions backed by extensive evidence and the review does a good job of suggesting several hypotheses and topics that merit further study.

Thanks.

Reviewer 2 Report

The review "Tau protein pathology in the brain after ischemia" is a very significant research topic in current medical scenario. The author has described the introduction part very well. Here they presented the effects of ischemic brain injury on neuronal damage and showed its connection with different diseases/dementia. Then they explained all the proteins involved in these diseases and how they related with 'tau' protein pathology. Following part of the review was mostly with tau proteins, its phosphorylation/dephosphorylation functions and its role in blood after ischemic injury. Here they missed the description of detailed structure of this protein.

The most important missing part here is, the schematic diagram of its structure, function and cross talk with other protein. At least one (or better more than one) model diagram must be present in a review paper, which is completely missing here. This makes the review unacceptable and hard for the reader.

Author Response

Reviewer 2. The review "Tau protein pathology in the brain after ischemia" is a very significant research topic in current medical scenario. The author has described the introduction part very well. Here they presented the effects of ischemic brain injury on neuronal damage and showed its connection with different diseases/dementia. Then they explained all the proteins involved in these diseases and how they related with 'tau' protein pathology. Following part of the review was mostly with tau proteins, its phosphorylation/dephosphorylation functions and its role in blood after ischemic injury. Here they missed the description of detailed structure of this protein.

Done. All changes in red.

The most important missing part here is the schematic diagram of its structure, function and cross talk with other protein. At least one (or better more than one) model diagram must be present in a review paper, which is completely missing here. This makes the review unacceptable and hard for the reader.

We added three figures about structure, function and cross talk.

Reviewer 3 Report

We read with great interest the review article “Tau protein pathology in the brain after ischemia” by Marzena Ułamek-Kozioł, Stanisław J. Czuczwar, Sławomir Januszewski and Ryszard Pluta

The review seeks to address two main ideas; first, the effect of ischemia-reperfusion brain damage on Tau protein (at both the genetic level and protein-structure level), second, the positive correlation between Tau protein dysfunction and the development of sporadic Alzheimer’s
disease type of neurodegeneration.  

Overall; the article is good and rich in references; however below are
comments about it, listed according to their respective order of appearance in the paper:

The article is descriptive and doesn't discuss the mechanism of phosphorylation and how it is triggered (the mechanist part is missing from the article)

-the Table used is extremely shallow and needs to be including information of animal model of ischemia,  tie points, species, etc....

o The title of the article is very brief and doesn’t reflect the exact purpose being addressed

o The abstract isn’t very specific and not well structured. The aim of the paper and the hypothesis suggested in lines 18 and 19 are not consequential thus using “therefore” is not appropriate. Besides that, several main ideas of the article were not included in the abstract.

o Line 98: in this section, the authors referred to other research papers to assess the changes in Tau protein gene expression; however, the experiment was not described in vivid details and the results are very brief.

o In the discussion section, the authors start it by referring to results and data of experiments not well explained and without using any graphs, which was misleading and not clear.

-the article would benefit from schematics and  figures describing the mechanism of tau phosphorylation

-tau proteolysis is not discussed at all bt proteases

o Line 254, Table 1 stands on its own without being mentioned in any previous section 

Author Response

Reviewer 3. We read with great interest the review article “Tau protein pathology in the brain after ischemia” by Marzena Ułamek-Kozioł, Stanisław J. Czuczwar, Sławomir Januszewski and Ryszard Pluta

The review seeks to address two main ideas; first, the effect of ischemia-reperfusion brain damage on Tau protein (at both the genetic level and protein-structure level), second, the positive correlation between Tau protein dysfunction and the development of sporadic Alzheimer’s disease type of neurodegeneration.  
 Overall; the article is good and rich in references; however below are comments about it, listed according to their respective order of appearance in the paper:

Thanks for useful comments. Changes in red.

The article is descriptive and doesn't discuss the mechanism of phosphorylation and how it is triggered (the mechanist part is missing from the article)

We presented this problem in Table 1.

-the Table used is extremely shallow and needs to be including information of animal model of ischemia, tie points, species, etc....

Done.

o The title of the article is very brief and doesn’t reflect the exact purpose being addressed

We changed title of article.

o The abstract isn’t very specific and not well structured. The aim of the paper and the hypothesis suggested in lines 18 and 19 are not consequential thus using “therefore” is not appropriate. Besides that, several main ideas of the article were not included in the abstract.

We changed abstract.

o Line 98: in this section, the authors referred to other research papers to assess the changes in Tau protein gene expression; however, the experiment was not described in vivid details and the results are very brief.

We added one new paper in this section and experiments were described in details.

o In the discussion section, the authors start it by referring to results and data of experiments not well explained and without using any graphs, which was misleading and not clear.

We added two figures in this section.

-the article would benefit from schematics and  figures describing the mechanism of tau phosphorylation

It is presented in Table 1.

-tau proteolysis is not discussed at all bt proteases

Done.

o Line 254, Table 1 stands on its own without being mentioned in any previous section 

We changed Table 1 and added it in text.

Round 2

Reviewer 3 Report

this is well organized article